# Forced and internal components of observed Arctic sea-ice changes

**Jakob Simon Dörr[1], David B. Bonan[2], Marius Årthun[1], Lea Svendsen[1], and Robert C. J. Wills[3,4]**

[1]Geophysical Institute, University of Bergen and Bjerknes Centre for Climate Research, Bergen, Norway
[2]California Institute of Technology, Pasadena, California, USA
[3]Department of Atmospheric Sciences, University of Washington, Seattle, Washington, USA
[4]Institute for Atmospheric and Climate Science, ETH Zurich, Zurich, Switzerland

**Correspondence:** Jakob Simon Dörr (jakob.dorr@uib.no)

**Abstract.** The Arctic sea-ice cover is strongly influenced by internal variability on decadal timescales, affecting both short-term trends and the timing of the first ice-free summer. Several mechanisms of variability have been proposed, but how these mechanisms manifest both spatially and temporally remains unclear. The relative contribution of internal variability to observed Arctic sea-ice changes also remains poorly quantified. Here, we use a novel technique called low-frequency component analysis to identify the dominant patterns of winter and summer decadal Arctic sea-ice variability in the satellite record. The identified patterns account for most of the observed regional sea-ice variability and trends, and they thus help to disentangle the role of forced and internal sea-ice changes over the satellite record. In particular, we identify a mode of decadal ocean–atmosphere–sea-ice variability, characterized by an anomalous atmospheric circulation over the central Arctic, that accounts for approximately 30 % of the accelerated decline in pan-Arctic summer sea-ice area between 2000 and 2012 but accounts for at most 10 % of the decline since 1979. For winter sea ice, we find that internal variability has dominated decadal trends in the Bering Sea but has contributed less to trends in the Barents and Kara seas. These results, which detail the first purely observation-based estimate of the contribution of internal variability to Arctic sea-ice trends, suggest a lower estimate of the contribution from internal variability than most model-based assessments.

## 1 Introduction

In response to increasing greenhouse gas concentrations, Arctic sea ice has declined in all seasons over the satellite record since 1979 (Onarheim et al., 2018; Stroeve and Notz, 2018). However, the decline is overlaid by substantial internal variability on interannual-to-decadal timescales, which can enhance or mask the long-term externally forced trends (Serreze et al., 2016). Internal variability is also a dominant source of uncertainty in projections of Arctic sea ice over the next few decades (Bonan et al., 2021a). To reduce the uncertainty in Arctic sea-ice projections, it is thus necessary to understand the mechanisms underpinning decadal Arctic sea-ice variability and how these mechanisms manifest in the observed sea-ice cover. Furthermore, in order to accurately quantify the sensitivity of sea ice to external forcing, it is necessary to assess the relative role of internal variability and the long-term response to global warming on recent trends in the observed sea-ice cover.

A number of mechanisms for decadal variability in Arctic sea-ice cover have been proposed for different seasons and regions. For example, trends in winter sea-ice cover have been attributed to changes in the Arctic Oscillation (AO) or the North Atlantic Oscillation (NAO), the leading modes of atmospheric circulation variability over the Arctic and North Atlantic, respectively (Thompson and Wallace, 1998; Deser et al., 2000; Wang and Ikeda, 2000). The NAO creates a dipole between sea ice in the Barents Sea and in the Labrador Sea, through changes in the advection of heat, moisture and sea ice (Hegyi and Taylor, 2017). The winter AO is also thought to be important for changes in Arctic sea ice in the following summer by altering the sea-ice motion and creat-

ing sea-ice thickness anomalies (Rigor et al., 2002; Park et al., 2018). Since the 2000s, however, the connection between the AO and both the winter and summer sea ice has weakened (Stroeve et al., 2011; Ogi et al., 2016; Park et al., 2018).

Other studies that focus on summer Arctic sea ice have suggested that decadal variability connected to higher geopotential over northern Greenland and the central Arctic has accelerated the decline in summer sea ice since 1979 through warming and moistening of the lower troposphere and decreased cloudiness (Ding et al., 2017, 2022; Wang et al., 2022). This anomalous atmospheric circulation has furthermore been shown to be related to sea-surface temperatures in the Pacific Ocean (Baxter et al., 2019). Decadal variability in the North Pacific is also thought to be connected to winter temperature variability as well as trends in the Arctic (Svendsen et al., 2018) and in particular to winter and spring sea-ice conditions in the Bering Sea (Yang et al., 2020). Additionally, low-frequency variability in both summer and winter sea ice is affected by ocean heat transport from the Atlantic and Pacific oceans into the Arctic Ocean (Zhang, 2015; Årthun et al., 2019). The combined influence of these different mechanisms shapes the decadal variability of the Arctic sea-ice cover. However, an attempt to identify and disentangle the leading modes of decadal variability in the relatively short satellite record of Arctic sea-ice concentration has not been made.

Understanding past Arctic sea-ice variability – and predicting its future – requires an understanding of the relative roles of internally and externally forced variability. In summer, it is estimated that around 30 %–50 % of the observed decline in total Arctic sea-ice area since 1979 is due to internal variability (Kay et al., 2011; Day et al., 2012; Ding et al., 2017). Estimates for different regions nevertheless vary substantially. For example, England et al. (2019) found that in summer internal variability strongly affects the Kara, Laptev and Beaufort seas and to a lesser extent the East Siberian Sea. In spring and winter, England et al. (2019) suggest that nearly all of the sea-ice trends in the North Atlantic are the result of internal variability. However, all of the estimates have so far relied to a large extent on a combination of observations and climate model experiments, such as single-model large ensembles that properly separate the forced signal from internal variability (e.g., Bonan et al., 2021a; Holland and Hunke, 2022) or so-called pacemaker experiments that isolate the impact of regional sea-surface temperature variability on Arctic sea ice (Meehl et al., 2018). Such estimates rely on the ability of climate models to accurately capture the forced signal and the correct range and mechanisms of internal variability, which are not precisely known. An analysis of the role of internal decadal variability on regional and seasonal Arctic sea-ice cover based purely on observations has not previously been performed, largely because it is difficult to separate internal variability from the long-term trend based on the short observational record.

Here, we aim to identify and analyze dominant patterns of decadal variability in the summer and winter Arctic sea-ice concentration over the satellite record using a recently developed method named low-frequency component analysis (LFCA; Wills et al., 2018), which is an objective method that identifies spatial anomaly patterns with the highest ratio of low-frequency variance to total variance. LFCA has been used to disentangle the long-term (forced) warming from decadal variability in sea-surface temperature data (Wills et al., 2018, 2020; Årthun et al., 2021b) and to identify sources of low-frequency variability in Antarctic sea ice (Bonan et al., 2023), which is a companion study. Using this method enables us to estimate the contribution of internal variability to observed regional decadal variability and trends over the last 4 decades and explore the atmospheric and oceanic mechanisms behind this variability. It is important to note that our goal is not primarily to find new mechanisms of Arctic sea-ice variability but rather to examine how each mode of variability manifests both spatially and temporally in the satellite record. Additionally, we use this method on a combination of sea-ice and sea-surface temperatures, where the forced response can be more easily identified. This allows us to provide the first purely observation-based estimate of the contribution of internal variability to Arctic sea-ice changes over the satellite record.

## 2  Materials and methods

We use gridded daily sea-ice concentration data from the Ocean and Sea Ice Satellite Application Facility (OSI SAF) (Lavergne et al., 2019) for the period 1979–2021. We compute seasonal averages for summer (July–September) and winter (January–March) for each year. We define summer and winter to be the 3 months leading up to the annual sea-ice minimum (in September) and maximum (in March), respectively. We perform the analysis for sea ice separately for summer and winter, on the native equal area $25\,km \times 25\,km$ grid.

To identify components of low-frequency variability, we use low-frequency component analysis (LFCA; Wills et al., 2018; Schneider and Held, 2001). LFCA isolates low-frequency variability by identifying low-frequency patterns (LFPs) and corresponding orthogonal low-frequency components (LFCs). The LFPs of a spatiotemporal data set are linear combinations of the leading empirical orthogonal functions (EOFs) that maximize the ratio of low-frequency variance to the total variance in the data set. Here, we determine low-frequency variance by applying a 10-year Lanczos filter for each grid point. We remove the linear trend before applying the filter, and we add it back to the filtered data afterwards. By taking into account both original and filtered data, the method identifies low-frequency modes with the minimum contribution from higher-frequency variability. The resulting LFPs are sorted by their ratio of low-frequency vari-

ance to total variance ($r$; hereafter called variance ratio), such that the leading modes describe low-frequency variability.

The two main parameters in LFCA are the number of EOFs retained and the cutoff period. Because we are focused on decadal variability, we set the cutoff period to 10 years using the filter explained above. We retain the six leading EOFs, which capture around 70 %–75 % of the total variance. If we increase the number of retained EOFs to 10 or decrease it to 3, the patterns are similar for summer sea ice. For winter sea ice, however, the second and third patterns change slightly, as some spatiotemporal features move from one pattern to the other. This is likely due to these two patterns having a similar timescale and ratio of low-frequency variance to total variance (for more details, see Sect. 3). The LFCA might therefore mix these two patterns. Furthermore, because of the long timescale of these patterns (around or more than 20 years), there are insufficient temporal degrees of freedom to properly separate them. We argue that retaining six EOFs is thus a good compromise to capture these two patterns and interpret them mechanistically.

We calculate the regional footprints of the identified low-frequency modes by projecting the LFPs and LFCs onto the sea-ice areas of different regions in the Arctic: the Barents and Kara seas; the Bering, Chukchi, and Beaufort seas; the East Siberian and Laptev seas (summer only); Baffin Bay and Labrador Sea (winter only); and the total Arctic (Fig. 1). To do this, we multiply each LFP by its corresponding LFC and sum over the target region. This results in time series of sea-ice area anomalies for each region that are associated with each LFP and LFC. We note that from January–March the Beaufort and Chukchi seas are fully ice covered, such that winter sea-ice variability in the Bering, Chukchi and Beaufort seas region is only occurring in the Bering Sea.

We assess physical mechanisms associated with the leading low-frequency modes by regressing sea-surface temperature (SST), 500 hPa geopotential height and surface winds from the ERA5 reanalysis from 1979–2021 (Hersbach et al., 2020) onto their LFCs. We also assess the association of the LFCs with common low-frequency climate indices such as the Pacific Decadal Oscillation (PDO; Mantua et al., 1997), Arctic Oscillation (AO; Thompson and Wallace, 1998) and North Atlantic Oscillation (NAO; Wallace and Gutzler, 1981) available from NOAA as well as the North Pacific Gyre Oscillation (NPGO) from Di Lorenzo et al. (2008). Further, to find a low-frequency mode which best represents the long-term ("forced") response of Arctic sea ice, we consider spatiotemporal variability in the atmosphere, ocean and sea ice together. This allows us to more robustly separate modes with a similar spatial footprint in sea-ice concentration (but different oceanic and atmospheric patterns) from the first LFP. To do so, we perform a combined LFCA with sea-ice concentration, Northern Hemisphere 500 hPa geopotential height and global SSTs. We interpolate the OSI SAF sea-ice concentration onto the same regular $1° \times 1°$ grid from ERA5. We normalize each field by the trace of its covari-

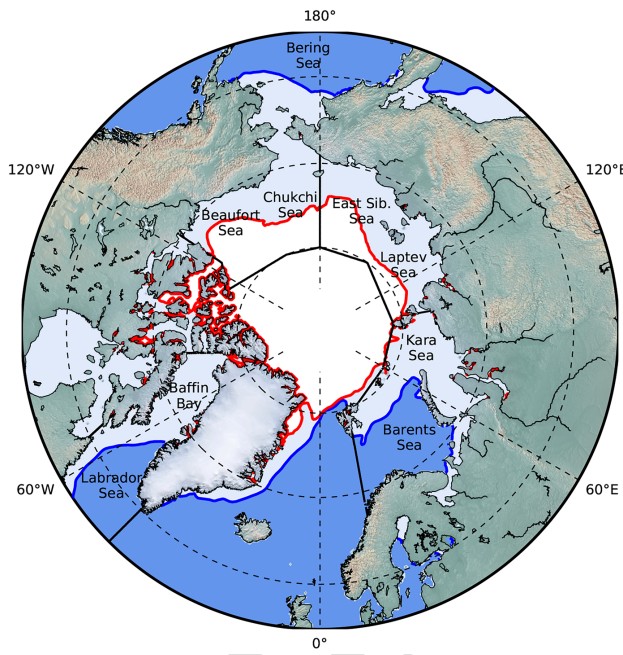

**Figure 1.** Map of the Arctic Ocean and its shelf seas. Regions used are bounded by solid black lines. The blue line represents the average observed winter (January–March) sea-ice cover from 1979–2021, and the red line is the average summer (July–September) sea-ice cover, both based on 50 % sea-ice concentration.

ance matrix and then flatten the two spatial dimensions of each data set into one dimension. We then concatenate the three fields along their spatial dimensions for each time step. The resulting LFPs take into account spatiotemporal variability in all three fields. This method is helpful for isolating anthropogenic signals in noisier climate fields. For more details, see Wills et al. (2020). We perform the combined analysis for both winter and summer sea ice. For winter, we use the January–March sea-ice concentration and geopotential height, as well as the winter-centered annual mean (July–June) SSTs. For summer, to assess an internal mode of decadal summer sea-ice variability suggested by, e.g., Ding et al. (2017), we use July–September sea-ice concentration, June–August geopotential height and annual mean (January–December) SSTs. We use annual mean instead of seasonal SSTs because this allows us to better separate internal modes from the global mean warming signal.

## 3 Results

### 3.1 Leading patterns of decadal sea-ice variability

We begin by performing LFCA on Arctic sea-ice concentration alone. The dominant low-frequency patterns (LFPs) and their corresponding time series (LFCs) are shown in Fig. 2. For summer (July–September, Fig. 2a–c), the first LFP is centered around the Chukchi, East Siberian and Laptev seas,

where observed trends in sea-ice concentration are largest. This pattern accounts for approximately 75 % of the spatiotemporal variability on long (> 10 years) timescales and has a relatively large low-frequency variance to total variance ratio ($r = 0.94$). The associated LFC shows a nearly monotonic decrease, which accelerates after the 1990s. This pattern can be interpreted as an Arctic-wide long-term ice loss mode. The second LFP features a tripole with negative anomalies in the Barents–Kara seas and the Beaufort Sea and positive anomalies in the Laptev and East Siberian seas. LFP2/LFC2 accounts for approximately 10 % of the low-frequency variance and has a smaller variance ratio ($r = 0.72$) than LFP1. The associated LFC shows pronounced decadal variability, particularly before 2000, with a timescale of around 15 years. The third pattern shows changes of the same sign throughout the Arctic Ocean, except for the Chukchi and Greenland seas. The corresponding LFC shows higher-frequency variability with a timescale of around 5 years, overlaid by low-frequency variations, including an upswing around the 2000s. The ratio of low-frequency variability is low ($r = 0.23$) for this pattern.

For winter (January–March, Fig. 2d–f), the first LFP shows a pan-Arctic sea-ice concentration signal with a high variance ratio, accounting for 51 % of the low-frequency variance. The associated LFC shows a long-term negative trend, accelerating after the 1990s, so it can be interpreted as a mean ice loss mode. The second pattern features a quadrupole on the Atlantic and Pacific sides. LFP2 accounts for approximately 20 % of the low-frequency variance and shows variability with a timescale of around 20 years ($r = 0.66$). The third pattern is dominant in the Bering Sea and accounts for approximately 10 % of total low-frequency variance. The associated LFC shows substantial low-frequency variability ($r = 0.52$), and it is positive for much of the 2000s, except for a strong reversal since 2010.

## 3.2 Decadal variations in regional sea-ice area

We next examine how the leading low-frequency patterns manifest in the regional sea-ice area (see Fig. 1) from 1979–2021 by projecting the LFPs and their LFCs onto the regional sea-ice area anomalies. We estimate the proportion of variance accounted for in the sea-ice areas by the different modes using the squared Pearson correlation coefficient. The evolution of the first mode follows the total Arctic sea-ice area closely for both summer and winter (Fig. 3a and b), accounting for around 95 % of the variability in the sea-ice area. This confirms that the first modes capture a pan-Arctic ice loss mode in both seasons. The close relation of LFC1 with the total sea-ice area either means that the sea-ice area is already a good indicator of the forced response of sea ice or that LFC1 includes decadal variability affecting the entire ice pack. We return to this issue in Sect. 3.4. For each region, the first mode generally captures less than 80 % of the variability, indicating that there is regional variability not

explained by the pan-Arctic loss mode. Indeed, large parts of the remaining variability are accounted for by LFC2 and LFC3, as indicated by the light colored variance bars with hatching (Fig. 3c–g).

For summer, the first LFP captures the long-term trend in all three regions. For the Barents–Kara region (Fig. 3c), the first mode accounts for approximately 60 % of the variability in sea-ice area (including the trend), while the third pattern accounts for approximately 10 % (30 % for the detrended data). For the Pacific sector (Fig. 3e), the first mode accounts for approximately 75 %. The remaining variability is largely accounted for by higher-frequency variability captured by the last three modes, particularly the fourth and fifth modes (Fig. A1a–c). For the East Siberian and Laptev seas (Fig. 3g), the first mode captures 70 % of the variability, and the second mode accounts for a large part of the remaining variability. It is important to note that there are large decadal variations in sea-ice area in this region before the 2000s, which are accounted for by the second mode, such as the large negative anomaly around 1990. This is consistent with Desmarais and Tremblay (2021), who find strong decadal sea-ice variability in the East Siberian Sea in summer.

For winter in the Barents and Kara seas (Fig. 3d), the first mode accounts for 70 % of the sea-ice area evolution. Another approximately 15 % is accounted for by the second mode (quadrupole), which also largely accounts for the deviations of the sea-ice area from the first mode. The second mode also accounts for parts of the decadal variability in the Labrador Sea, particularly between 1979 and 2000 (Fig. 3h). For the Pacific sector (Fig. 3f), the first mode accounts for almost none of the variability in the sea-ice area. Instead, most sea-ice variability is accounted for by the third mode, with 80 % variance explained. The third mode captures the slow increase in sea-ice area and the pronounced positive anomalies around 2010, followed by a strong decrease thereafter. This is an indication that the slow increase in sea-ice area in the Pacific sector of the Arctic was likely a result of internal variability, consistent with Svendsen et al. (2021).

## 3.3 Mechanisms of decadal sea-ice variability

Assuming for now that the first LFP represents the long-term forced signal (discussed in Sect. 3.4), we next look at mechanisms underpinning the second and third LFPs. We do so by regressing (January–March for winter, July–September for summer) sea-surface temperature (SST), 500 hPa geopotential height and surface winds from ERA5 onto the LFCs for zero lag in Fig. 4 and leading the LFCs by 6 months in Fig. 5. We choose a 6-month lag to assess possible summer-to-winter and winter-to-summer linkages in sea ice (e.g., Rigor et al., 2002).

The second LFP in summer (tripole) is only weakly connected to SST and atmospheric circulation variability in summer, except for SST close to the sea-ice edge (Fig. 4a). A notable feature is seen over the Bering Sea, with anoma-

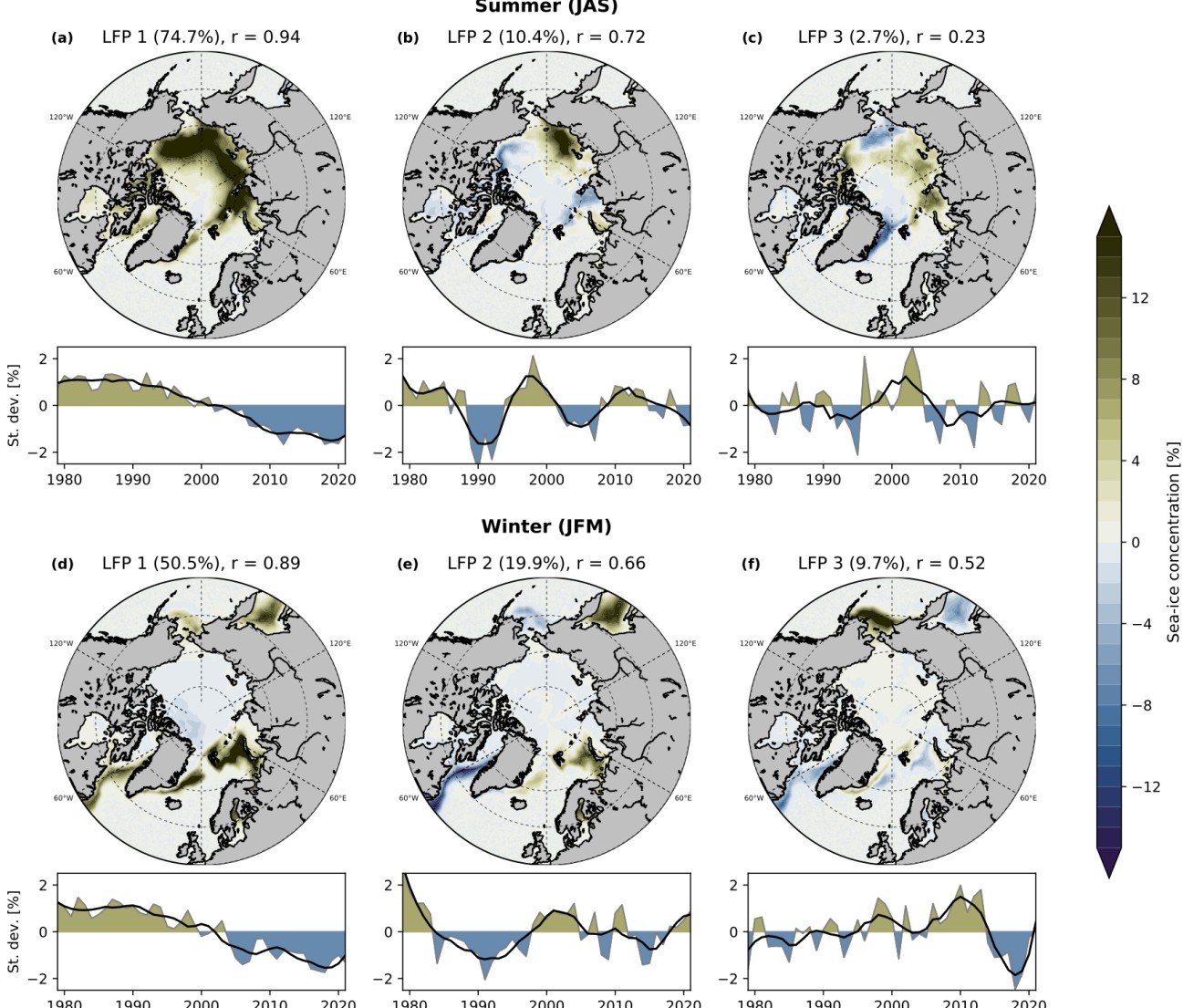

**Figure 2.** The three leading low-frequency patterns (maps, color bar) and their associated time series (line plots) for summer (July–September, **a–c**) and winter (January–March, **d–f**) Arctic sea-ice concentration. The fraction of explained low-frequency variance (in %) and the ratio *r* of low-frequency variance to total variance is given for each pattern. TS1

lous meridional winds and negative SST, along with positive geopotential anomalies over the Aleutian Islands. Much more notable is the connection to circulation anomalies over the preceding winter (Fig. 5a), with higher geopotential over much of the central Arctic and Greenland, lower geopotential over the midlatitudes, and anomalous transpolar winds towards the Siberian coast. The regression pattern is somewhat reminiscent of the Arctic Oscillation (AO), and the second summer LFC is significantly correlated with the winter-centered (July–June) annual mean AO index (correlation coefficient $R = -0.61$). The winter AO is known to influence sea ice in the following summer through sea-ice motion and subsequent thinning (Rigor et al., 2002; Park et al., 2018; Brunette et al., 2019; Gregory et al., 2022), a connection

which was particularly strong in the 1990s and 2000s but has since weakened (Stroeve et al., 2011; Ogi et al., 2016). Anomalous transpolar winds may also explain the dipole between the Beaufort and the East Siberian seas (Fig. 2b). Our results suggest that the second LFP captures the influence of the winter AO on summer sea ice.

The third LFP in summer is connected to a dipole in geopotential height over western Greenland and the Canadian Arctic Archipelago on the one side and the Nordic Seas on the other side (Fig. 4b), which sets up winds along the transpolar drift between the Siberian coast and the Greenland Sea. This pattern is reminiscent of the Arctic dipole pattern (Overland and Wang, 2005; Choi et al., 2019). There are some weak positive SST anomalies over the North Pa-

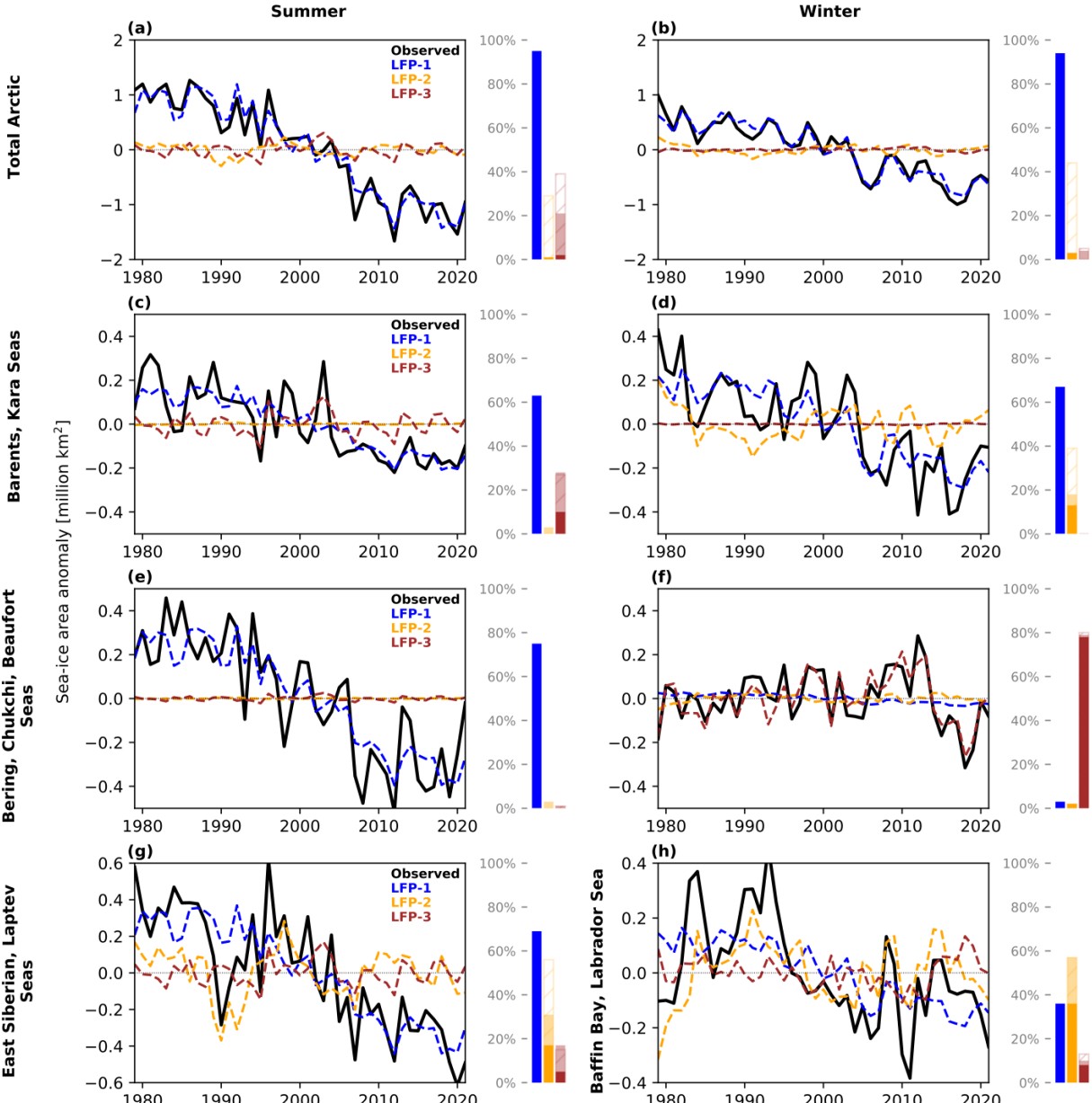

**Figure 3.** Observed Arctic sea-ice area anomalies (in $\times 10^6$ km$^2$) (black lines), computed from the first (blue), second (orange) and third (brown) low-frequency modes for the total Arctic **(a, b)**, Barents–Kara seas **(c, d)**, and the Bering–Chukchi seas **(e, f)** for summer (left column) and winter (right column), as well as East Siberian–Laptev seas in summer **(g)** and Baffin Bay and Labrador Sea in winter **(h)**. Bars indicate the proportion of the variance ($R^2$), where $R$ is the Pearson correlation coefficient, in the raw sea-ice area (SIA) time series that is accounted for by each low-frequency mode. The dark color denotes the explained variance in the raw time series, the light color denotes the explained variance in the detrended SIA time series, and the hatching denotes the explained variance in the difference between the raw SIA and the SIA computed from the first low-frequency mode.

cific connected to this pattern. The high geopotential over the Nordic Seas is also seen in the previous winter (Fig. 5b), along with lower geopotential over western Siberia and the Azores, suggesting possible preconditioning through winter circulation anomalies. Furthermore, in the previous winter, the SST pattern is reminiscent of the negative phase of the El Niño–Southern Oscillation (ENSO), and the atmospheric cir-

culation is reminiscent of the Pacific–North America (PNA) pattern. This is consistent with literature showing a possible role of ENSO in the record-low sea-ice extent in 2012 (Jeong et al., 2022) and a connection of the 2007 sea-ice minimum with the PNA (L'Heureux et al., 2008). Both minima are captured by LFC3 (Fig. 2c).

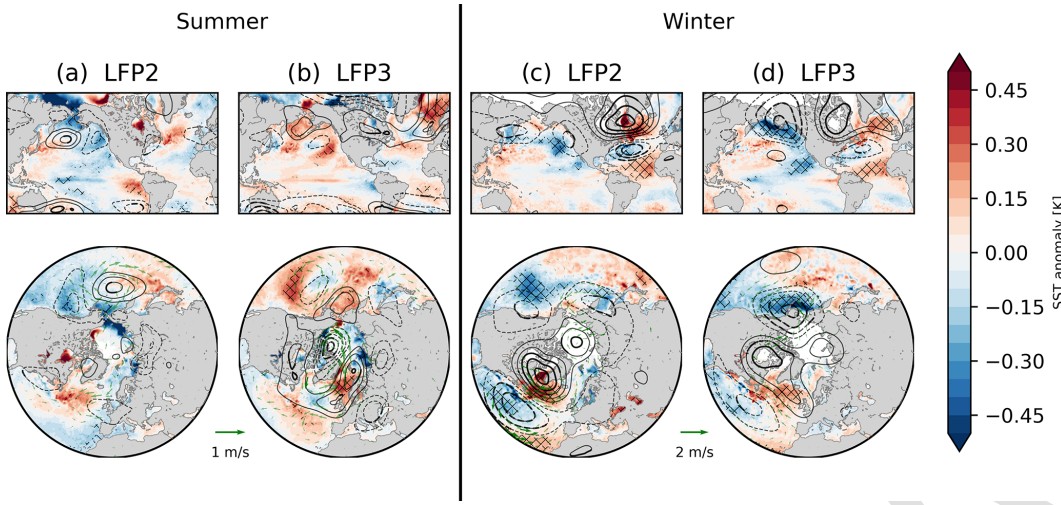

**Figure 4.** Regression coefficients of the second and third low-frequency components (sea-ice concentration, 10-year cutoff, six EOFs retained) and detrended seasonal mean sea-surface temperature (color) and geopotential height (contours; dashed lines show negative values; contour interval: 60 m) for summer (July–September, **a, b**) and winter (January–March, **c, d**), using two different projections (top and bottom row). Green arrows in the bottom row denote the regression of seasonal mean surface winds on the low-frequency components. Thick arrows, hatching and thick contour lines denote significant values at the 95 % confidence interval.

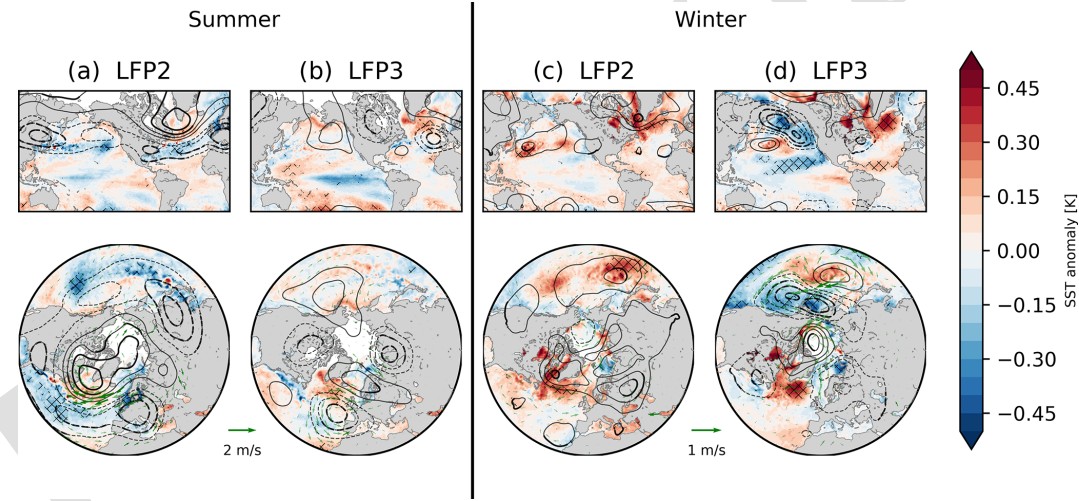

**Figure 5.** Same as Fig. 4 but with all fields leading the LFCs / sea-ice anomalies by 6 months.

For winter, the second LFP is connected to higher geopotential over the Arctic (mainly over the central Arctic Ocean, Greenland and northern Canada) and lower geopotential over the midlatitudes (Fig. 4d). Similarly to the second summer LFP, this pattern is reminiscent of the AO or NAO, and the correlation with the winter-centered annual AO index is significant ($R = -0.4$). The associated LFC is also weakly correlated with the LFC associated with the second summer LFP ($R = 0.31$). The associated winter SST pattern shows the strongest signal south of Greenland, co-located with the atmospheric circulation anomaly. A similar SST pattern does not appear for the previous summer (Fig. 5d), which hints at seasonal atmospheric circulation anomalies likely being the main driver for this mode. This mode is consistent with a proposed connection and interaction of the AO/NAO with winter sea-ice variability in the Atlantic sector (Deser et al., 2000; Strong et al., 2009).

The third winter LFP, which has the largest footprint in the Bering Sea (Figs. 2f and 3f), is associated with a stronger Aleutian Low, along with a wave train over North America reminiscent of the PNA, with positive geopotential anomalies over northern Canada and negative anomalies over the Gulf Stream (Fig. 4c). The associated SST anomaly pattern is reminiscent of the Pacific Decadal Oscillation (PDO) or the North Pacific Gyre Oscillation (NPGO; Di Lorenzo et al., 2008), both for winter and the preceding summer

(Figs. 4c and 5c). LFC3 is correlated with both the winter-centered annual mean PDO index ($R = -0.4$) and NPGO index ($R = 0.42$). This is consistent with Yang et al. (2020), who suggested that decadal spring Bering Sea ice extent variability is connected with North Pacific SST variability. Sea ice in the Bering Sea has also shown large decadal variability in winter in recent decades (Fig. 3f), which is captured by LPF3. The third winter LFP might thus isolate this mode of covariability of Bering Sea ice and North Pacific SST variability. SST anomalies associated with LFC3 also extend into the Atlantic Ocean, with anomalies over the Gulf Stream in phase with those in the northeast Pacific. The negative SST anomalies over the North Pacific are still seen in the previous summer, which further suggests a Pacific origin of the circulation anomalies.

### 3.4   Isolating the impact of external forcing

For both summer and winter, we have identified patterns of pan-Arctic sea-ice loss in the first LFPs (Fig. 2a and d). These patterns were separated from the other patterns due to their pan-Arctic nature and the high ratio of low-frequency variance to total variance. It is therefore tempting to interpret these patterns as the impact of rising global temperatures on the Arctic sea ice as a whole, i.e., the forced signal from climate change. However, these patterns essentially follow the evolution of the total Arctic sea-ice area, which itself is influenced by internal variability on interannual and decadal timescales (Swart et al., 2015; Ding et al., 2017). This is especially the case for summer, where the remaining sea-ice cover is limited to the central parts of the Arctic Ocean (Fig. 1).

The first LFC for summer shows some decadal variability on top of the long-term decrease, especially between 2000 and 2012 (Fig. 2a). The first LFP could thus be a combination of the forced signal and internal variability. To circumvent this mixing of spatially similar patterns, we perform a combined LFCA with sea-ice concentration in July–September together with Northern Hemisphere 500 hPa geopotential in June–August and global annual mean SST from ERA5 (see "Materials and methods" section). We use annual SSTs as this choice better separates the LFCs, but note that using seasonal SSTs does not qualitatively change our results. We use June–August for geopotential height because the summer sea-ice area (and extent) is thought to be connected to atmospheric circulation anomalies over the central Arctic in early summer (Wettstein and Deser, 2014; Ding et al., 2017).

The first three patterns of each variable of the combined analysis in summer are shown in Fig. 6. The first combined LFP shows a pan-Arctic sea-ice decrease, along with a general geopotential height increase over the Northern Hemisphere, except over an area south of Greenland. The strongest geopotential height increases are over Greenland, the central Arctic and eastern Europe. The SST pattern also shows a general increase, except for the area south of Greenland

(often referred to as the "Atlantic warming hole"; Keil et al., 2020), the Pacific upwelling region west of South America and in the Pacific sector of the Southern Ocean. This first combined LFP is very similar to the spatial patterns of the linear trends in sea ice, geopotential height and SST in ERA5 (not shown). It is furthermore consistent with estimates of the global warming pattern since 1979 using similar pattern analysis techniques (Wills et al., 2020), suggesting that this mode is a good approximation of the forced response. Compared to the first ice-only pattern, the combined LFC1 shows a much smoother and more linear evolution, especially after 2000. Two episodes of sea-ice increase can be seen after 1982 and 1991. These periods correspond to the last major volcanic eruptions (Mt. El Chichón and Mt. Pinatubo), which have been shown to have had a multiyear impact on Arctic sea ice (Gagné et al., 2017; Pauling et al., 2021). It is possible that the combined LFCA detects the impact of these eruptions.

The second combined pattern also features widespread negative anomalies in Arctic sea-ice concentration and positive anomalies in geopotential height over the central Arctic. However, these positive anomalies are surrounded by lower geopotential heights over Alaska, Siberia and western Europe. The connection between low summer sea ice and anticyclonic circulation over the central Arctic has been discussed in previous literature (Ding et al., 2017, 2019, 2022). The SST pattern is reminiscent of the Interdecadal Pacific Oscillation (IPO; Henley et al., 2015). Cooler Pacific SSTs have been connected to reduced summer sea-ice extent, particularly before the 2012 record low (Meehl et al., 2018; Baxter et al., 2019; Screen and Deser, 2019; Jeong et al., 2022). Similar to the SST regression pattern of the third ice-only winter mode (Figs. 4d and 5d), the SST anomalies in the North Atlantic south of Greenland are opposite of the tropical Pacific anomalies. Interestingly, the amplitude of the second pattern seems to have increased since the 2000s, which might indicate that summer sea ice is more influenced by atmospheric variability, as it becomes thinner and younger (Sumata et al., 2023). However, an increasing atmospheric influence cannot be readily identified with LFCA because it assumes that the relative amplitudes of sea-ice, atmospheric and SST anomalies are constant in time. Finally, the third pattern is very similar to the second summer LFP for sea ice only (Fig. 2b), providing confidence that this represents a physical mode of variability.

The first and second patterns in the combined analysis share both the pan-Arctic sea-ice signal and the atmospheric circulation signal over the central Arctic. Because of this spatial similarity, it is likely that the first pattern from the ice-only analysis (Fig. 2a) is a mix of these two patterns. If we assume that the leading combined LFP is a cleaner representation of the global mean (forced) changes over the analysis period, we can estimate the impact of the second combined pattern as an additional internal mode affecting the entire summer Arctic sea ice. We again project the pan-Arctic and regional sea-ice area of the three joint modes and com-

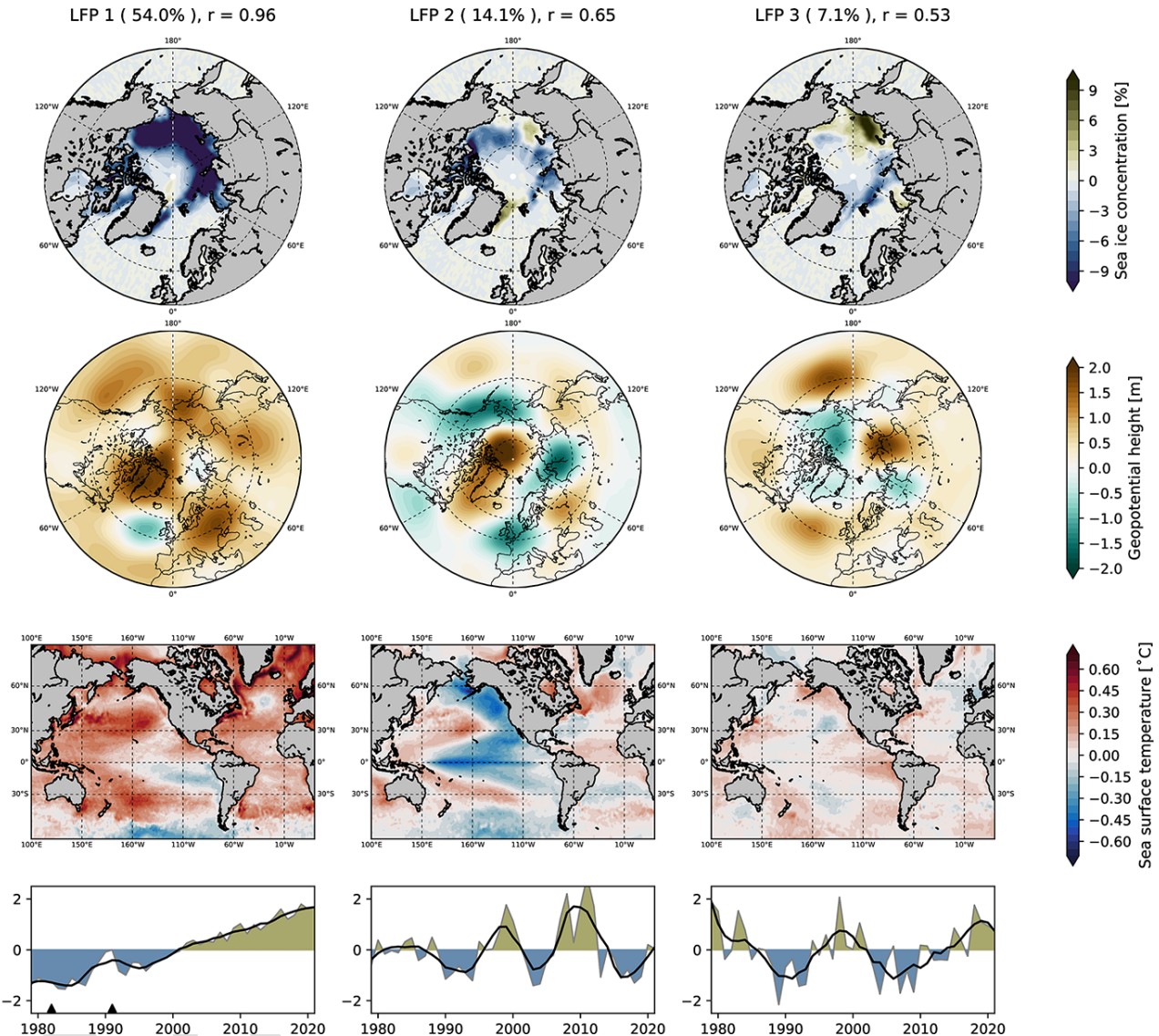

**Figure 6.** The three leading low-frequency patterns (maps) and their associated time series (line plots) using summer (July–September) sea-ice concentration and early summer (June–August) 500 hPa geopotential and annual mean (January–December) sea-surface temperatures. The fraction of explained low-frequency variance (in %) and the ratio of low-frequency variance to total variance are given for each pattern. Black triangles in the bottom left time series indicate the two major volcanic eruptions of Mt. El Chichón in 1982 and Mt. Pinatubo in 1991.

pare them with the raw sea-ice area in Fig. 7. The first mode accounts for approximately 90 % of the variability in the total Arctic summer sea-ice area, implying that around 10 % of the variability is internally driven. However, the contribution of internal variability on sea-ice trends over shorter periods within the observational record can be higher. Here, the second pattern accounts for some notable periods of faster and slower pan-Arctic summer sea-ice decline, which have been discussed in previous studies (Ding et al., 2022; Wang et al., 2022). For example, sea-ice loss accelerated substantially between 2000 and 2012, a feature that is captured by the second mode. In fact, around 30 % of the trend from 2000–2012 can

be accounted for by this mode, while the first mode accounts for just over 50 % in that period.

We summarize the estimates of the forced and internal contribution to trends over different periods and in the different regions in Fig. 8. If the first mode is an accurate representation of the long-term Arctic sea-ice response to increasing temperatures, the impact of internal variability to trends in summer Arctic sea-ice area in 2000–2012 would be approximately 40 %–50 % and would be approximately 10 % for the period 1979–2012 (Fig. 8a). These estimates are lower than previous estimates of more than 50 % for 2000–2012 and around 30 %–50 % for 1979–2010 (Ding et al., 2019, 2022). The discrepancy in estimates might relate to the fact that

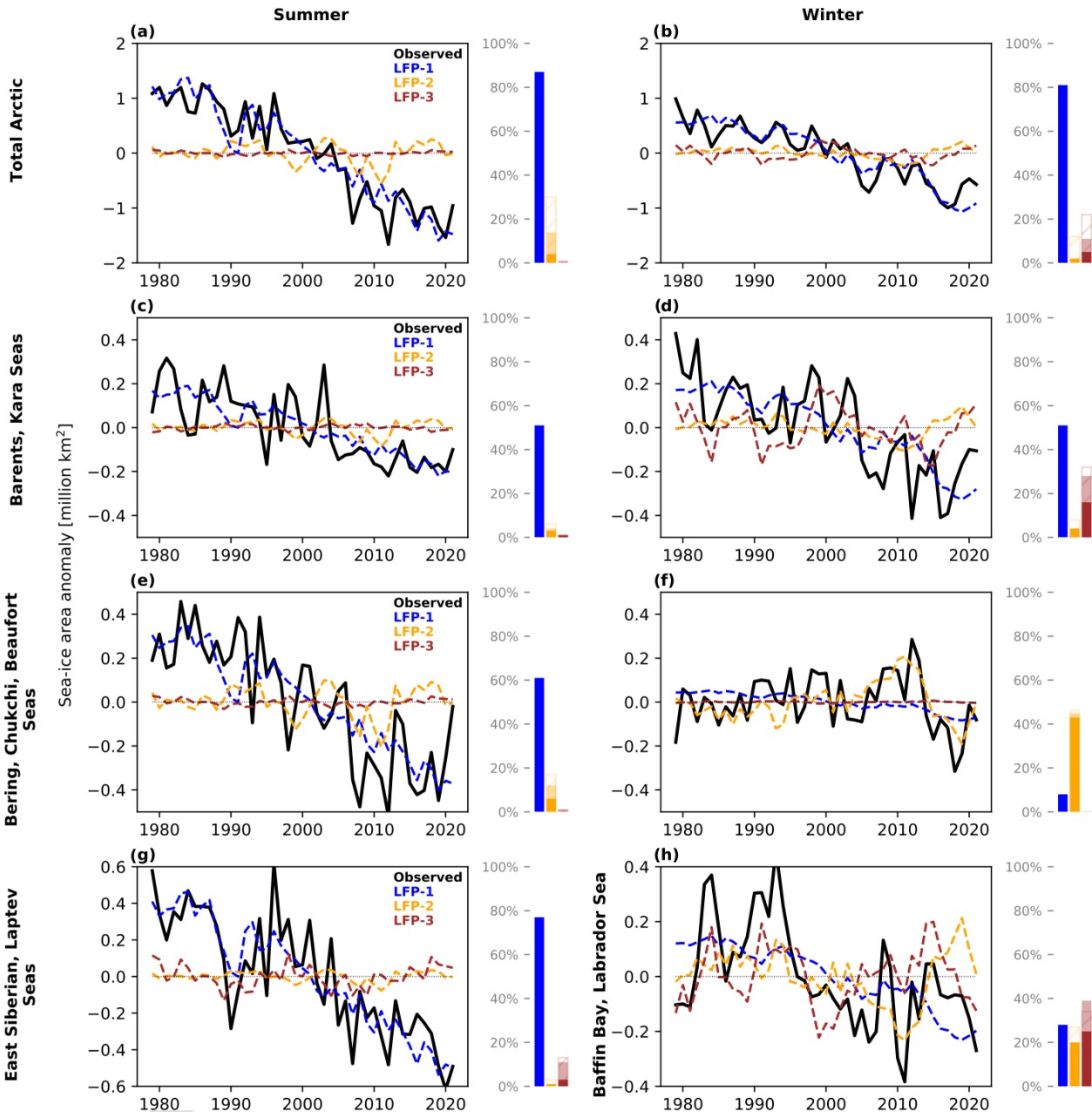

**Figure 7.** Observed Arctic sea-ice area (SIA) anomalies (in $\times 10^6$ km$^2$) (black lines) computed from the first (blue), second (orange) and third (brown) combined low-frequency modes, using sea-ice concentration, geopotential height and SSTs. Time series are shown for the total Arctic **(a, b)**, Barents–Kara seas **(c, d)**, and the Bering–Chukchi seas **(e, f)** for summer (left column) and winter (right column), as well as East Siberian–Laptev seas in summer **(g)** and Baffin Bay and Labrador Sea in winter **(h)**. Bars indicate the proportion of the variance ($R^2$), where $R$ is the Pearson correlation coefficient, in the raw SIA time series that is accounted for by each low-frequency mode. The solid color denotes the explained variance in the raw time series, the light color denotes the explained variance in the detrended SIA time series, and the hatching denotes the explained variance in the difference between the raw SIA and the SIA computed from the first low-frequency mode.

this analysis is based purely on observations, and previous analyses use a combination of models and observations. This would thus imply that the sensitivity of summer sea ice to external forcing is higher than previously thought. We find only a small contribution of internal variability for trends over the full period (1979–2021). Our results also suggest that

the recent slowdown in summer sea-ice loss between 2012 and 2021 is internally driven by the second combined mode (Fig. 7a, c and e), which is associated with high geopotential over the central Arctic and Pacific Ocean variability (Fig. 6).

We repeat the trend analysis for summer for the different subregions of the Arctic and find that for the 2000–2012

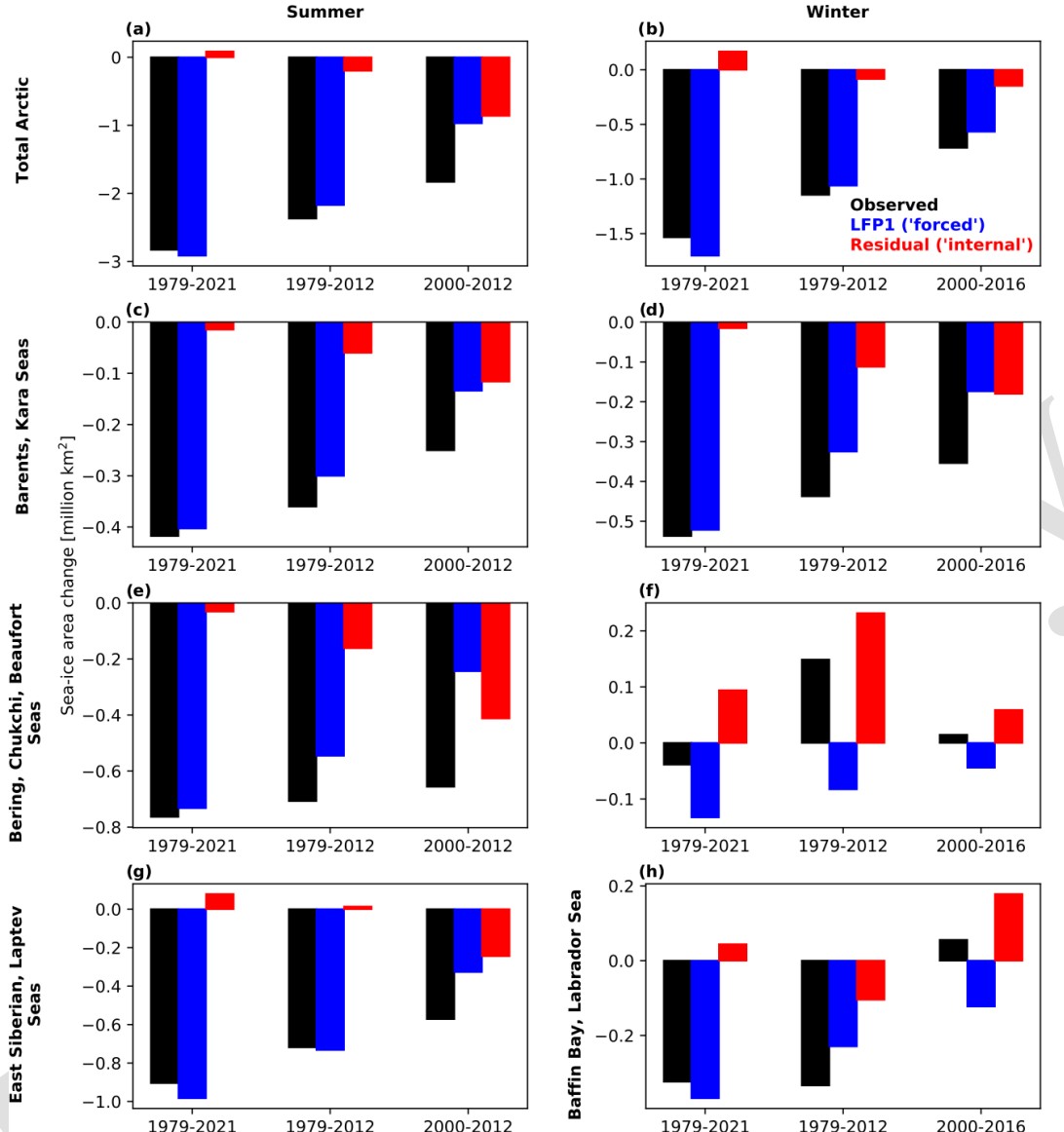

**Figure 8.** Estimates of forced (LFP1; blue bars) and internal (residual: total trend minus LFP1; red bars) contributions to trends in regional sea-ice area computed from the first combined low-frequency mode using sea-ice concentration, geopotential height and SSTs. The observed (total) trends (in $\times 10^6$ km$^2$) that change over the period are shown by the black bars. Trends are shown for the total Arctic **(a, b)**, Barents–Kara seas **(c, d)** and the Bering–Chukchi seas **(e, f)** for summer (left column) and winter (right column), as well as East Siberian–Laptev seas in summer **(g)** and Baffin Bay and Labrador Sea in winter **(h)**.

and 1979–2012 periods the contribution of internal variability of the Bering–Chukchi–Beaufort and the Barents–Kara sea-ice area is slightly higher than for the pan-Arctic sea-ice area. The internal contribution is notably lower for the East Siberian and Laptev seas, which is consistent with England et al. (2019).

We also repeat the combined analysis with winter (January–March) sea-ice concentration and geopotential height, as well as winter-centered annual mean (July–June) SSTs (Fig. A2), but the patterns are similar to those in Fig. 2d–f, except that LFP2 and LFP3 switch positions. This

suggests the first LFP in Fig. 2d might already be a good approximation of the forced response of winter sea ice. Projecting the combined modes onto regional sea-ice areas (Fig. 7b, d, f and h), we find that, similar to summer, the first mode captures approximately 80 % of the decadal variability in pan-Arctic winter sea-ice area. We estimate that around 10 % of the trend in total winter Arctic sea-ice area in 1979–2012 was internally driven and that internal variability might have decreased the total sea-ice loss over the full period (1979–2021) by around 10 % (Fig. 8b). Decadal trends in pan-Arctic and regional winter sea ice carry a higher footprint of internal

variability. For example, our trend analysis shows that up to 50 % of the accelerated decline in Barents–Kara winter sea ice between 2000 and 2016 was internally driven (Fig. 8d). The combined analysis also confirms that the evolution of the Bering Sea winter sea-ice area has so far been dominated by internal variability, which has favored an increase in sea-ice area. Nevertheless, just like in summer, our analysis suggests a smaller contribution of internal variability to observed trends and thus a higher sensitivity of winter sea ice to external forcing than previous model-based estimates, especially for the Barents and Kara seas, where it was assumed that internal variability has so far dominated winter and spring sea-ice trends (Onarheim and Årthun, 2017; England et al., 2019).

The leading SST patterns in Figs. 6 and A2 capture cooling in the Southern Ocean and the eastern tropical Pacific, which is also seen in temperature trends over the last decades. This could be a sign of a slow mode of internal variability being mixed into the patterns. However, the cooling in these regions might also be due to forced changes in the tropical Pacific and Southern Ocean (Seager et al., 2019; Wills et al., 2022; Heede and Fedorov, 2023) and thus part of the forced response. The regions of cooling (and reduced warming) in the first pattern coincide with regions of cooling in the second pattern in Fig. 6, and both patterns also feature high early-summer geopotential height over Greenland and the central Arctic, which is suggested to favor summer sea-ice loss (Wettstein and Deser, 2014; Ding et al., 2017; Wang et al., 2022). Furthermore, LFCA tries to maximize the fraction of the trend over the full period captured by LFC1, which favors a higher forced fraction for the full time period in particular. Thus, if internal variability is mixed into the first pattern, it would likely be connected to increased summer sea-ice loss, and our estimates of the contribution of internal variability to accelerated summer sea-ice loss might represent a lower limit.

There is also uncertainty in the identified patterns of long-term Arctic sea-ice loss. Slow modes of internal variability with a timescale of 25 years or more cannot be perfectly separated by LFCA in such a short observational record and might be mixed into the leading pattern, even for the combined analysis. To understand the limits of LFCA in isolating the forced signal in satellite-era sea-ice changes, we tested the ability of LFCA to separate a forced signal by applying the same combined analysis from Fig. 6 to 40 simulations of the Community Earth System Model Version 1 Large Ensemble (CESM1-LE; Kay et al., 2015) over 1979–2021. LFCA is able to extract similar global mean patterns from all individual members, and the reconstructed summer sea-ice area is highly correlated with the ensemble mean sea-ice area which can be interpreted as the forced signal (Fig. A3). Based on this, we can reject the null hypothesis that LFP1 is unrelated to the forced signal in CESM1-LE or in the satellite observations. Nevertheless, there are still differences in interdecadal trends between the members' first LFP and the

ensemble mean (i.e., the forced response of CESM1-LE), which can be attributed to internal low-frequency modes that get mixed with the forced trend in the LFCA (Deser and Phillips, 2023). Trends in the ensemble mean geopotential height in CESM1-LE are also much more spatially uniform than in any of the first LFP from the single members, although the ensemble mean shows larger trends over high latitudes compared to lower latitudes (Fig. A4). This further indicates that regional features (e.g., the high-pressure center over Greenland; Fig. 6) may be a result of internal variability being mixed into the first pattern.

## 4   Discussion and conclusions

Internal variability of the climate system influences decadal trends in the Arctic sea-ice cover and can mask the response of sea ice to global warming. To improve decadal predictions as well as long-term projections, it is important to understand the different modes of decadal variability that have influenced the Arctic sea-ice cover over the last few decades. In this study, we used low-frequency component analysis (LFCA) to separate decadal modes of variability from the observational record of summer and winter Arctic sea-ice concentration. The modes separated by LFCA account for most of the observed decadal sea-ice variations in different regions of the Arctic. We showed that the patterns can be related to mechanisms of atmospheric and oceanic variability discussed in the literature.

For summer, we identify a mode that captures the response of summer sea ice to atmospheric circulation anomalies during the previous winter, often quantified by the Arctic Oscillation, consistent with previous literature (Rigor et al., 2002; Gregory et al., 2022). For winter, we find a quadrupole mode which is also connected to Arctic Oscillation-like anomalies, but the connection is weaker. This mode is similar to the quadrupole mode discussed by Close et al. (2017) but with less focus on the Bering Sea. Although we find only a weak relationship of this mode with the Arctic Oscillation and the North Atlantic Oscillation (NAO), the spatial pattern is similar to the mechanism described by Luo et al. (2017), where a positive NAO and high pressure over northeastern Europe and Siberia act together to influence sea ice over the Barents–Kara seas. We further find a mode in winter connected to decadal variability in the Aleutian Low and SSTs in the Pacific which, consistent with Yang et al. (2020), captures sea-ice variability in the Bering Sea. This method is able to separate each of these mechanisms, explain how they relate to the long-term sea-ice loss, and quantify their contributions to regional sea-ice variability and trends.

Even though the identified patterns seem to be connected to physical mechanisms identified previously, we acknowledge that our results and conclusions are based on a short (43 years) observational record. Low-frequency sea-ice variability with a period of more than 20 years, forced by, for

example, Atlantic Multidecadal Variability (Day et al., 2012; Zhang, 2015), will therefore be difficult to quantify in this analysis. The results may therefore overestimate the forced contribution to sea-ice changes and underestimate the internal contribution. One might expect the identified patterns and the results to become more robust in the future when the satellite record becomes longer. However, September sea ice is projected to disappear in the coming decades (Notz and SIMIP community, 2020; Årthun et al., 2021a; Bonan et al., 2021b). Furthermore, the winter ice edge will retreat to different regions (Årthun et al., 2021a), where mechanisms and modes of decadal variability might be different. Patterns of low-frequency variability might therefore not be stationary in time (Dörr et al., 2021).

LFCA is based on principal component analysis, which maximizes the spatiotemporal variance explained by modes. Therefore, more localized impacts such as the impact of ocean heat transport on sea ice are likely not captured by the method. For winter, there is a strong connection between Atlantic heat transport and sea ice in the Barents and Kara seas (Årthun et al., 2012), and low-frequency variability in summer sea ice is also connected to ocean heat transport (Zhang, 2015). We do note a connection between the sixth winter pattern (shown in Fig. A1f) and ocean heat transport through the Barents Sea Opening (not shown). It might thus be that wind-driven oceanic heat transport into the Barents Sea is part of this higher-frequency pattern, consistent with atmospherically driven ocean heat transport anomalies being more important on shorter timescales than on decadal timescales (Årthun et al., 2019; Lien et al., 2017). The fact that the combined analysis for winter captures similar patterns (Fig. A2) is also an indication that the addition of SST and atmospheric variability does not capture the influence of ocean heat transport. We acknowledge, thus, that the impact of Atlantic heat transport on low-frequency sea-ice variability is not fully captured by our method; therefore, our winter results might not fully account for the role of internal variability on winter sea ice.

To more accurately estimate the contributions of the forced signal and internal variability to sea-ice trends, we used LFCA on a combination of sea-ice, oceanic and atmospheric variables. For winter, the combined analysis suggests that 80 % of decadal variability, as well as approximately 90 % of the sea-ice loss from 1979–2012, is accounted for by the forced mode. However, the internal variability is dominating sea-ice trends in the Bering Sea and accounting for more than half of the sea-ice loss in the Barents and Kara seas from 2000–2016, showing that the role of internal variability on decadal sea-ice trends is large. In summer, we were able to separate an additional pan-Arctic mode of variability connected to high geopotential over the central Arctic and Pacific Ocean variability, consistent with previous literature (Wettstein and Deser, 2014; Ding et al., 2017; Baxter et al., 2019). This led to an improved estimate of the long-term summer Arctic sea-ice loss. Based on these combined modes,

we estimate the contribution of internal variability to the period of accelerated summer sea-ice loss from 2000–2012 to be around 40 %–50 %. However, for the period of 1979–2012 the internal variability contribution is only about 10 %. These numbers represent the first estimate based purely on observations and are lower than previous estimates which have relied on a combination of climate models and observations. We note, however, that there is substantial uncertainty around these numbers due to the short observational record and the fact that model simulations show substantial multidecadal internal variability in Arctic sea-ice cover (Serreze et al., 2016), not all of which can be distinguished from the forced response using LFCA.

This study improves our understanding of decadal Arctic sea-ice variability and its role in the observed long-term decline of Arctic sea ice. A key result from this study is the ability to partition decadal sea-ice variability into distinct modes, which could be useful for predicting regional sea ice in the Arctic. For example, we find that the absence of strong summer sea-ice loss over the last 15 years is likely internally driven and related to Pacific variability (LFP2, Fig. 6; Screen and Deser, 2019; Baxter et al., 2019; Ding et al., 2019). Assuming this internal mode switches to its opposite phase, we expect it to contribute to accelerated summer Arctic sea-ice loss over the next decade.

## Appendix A

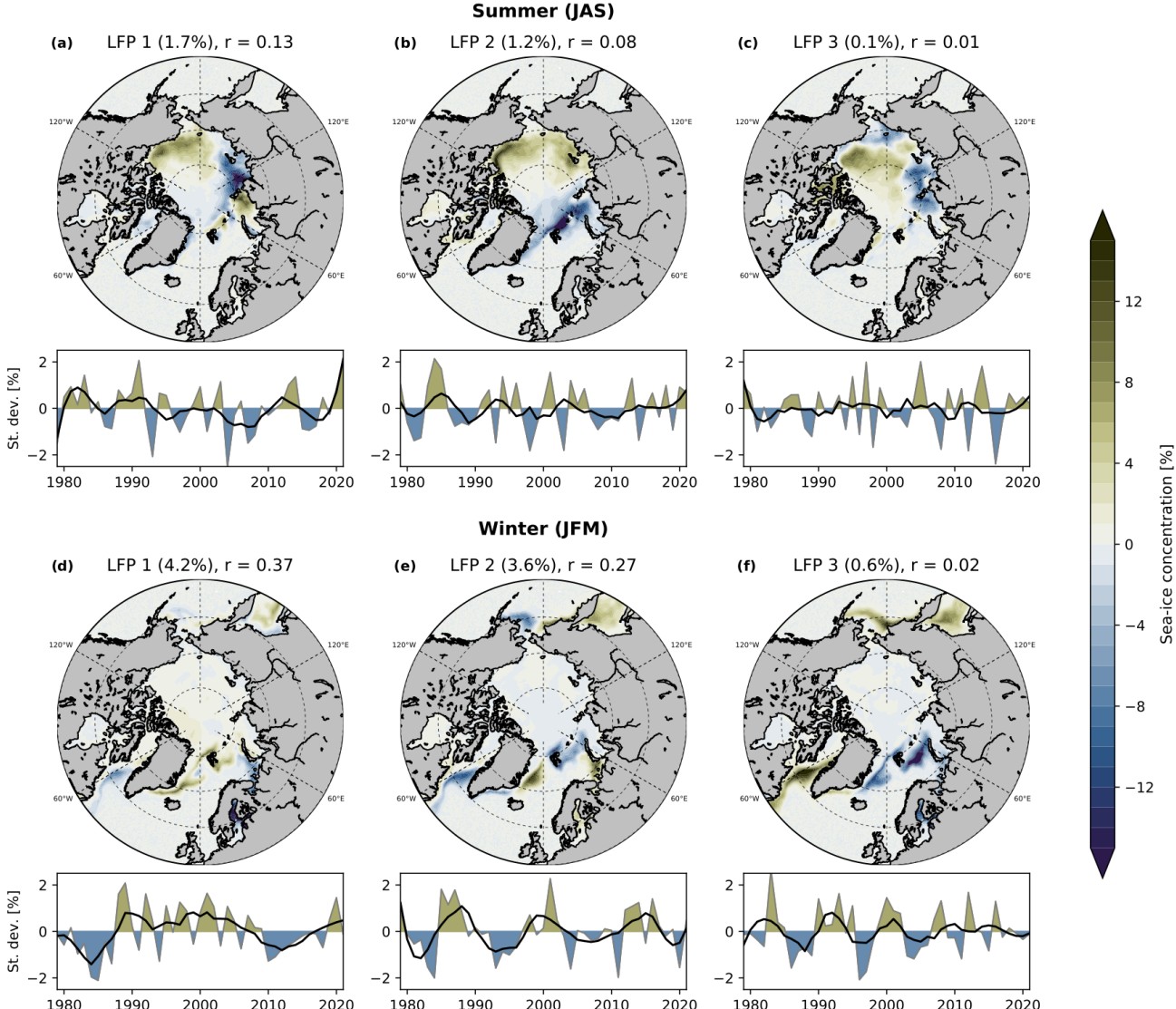

**Figure A1.** Fourth, fifth and sixth low-frequency patterns (maps) and their component time series (line plots) for summer (July–September, **a–c**) and winter (January–March, **d–f**), using a 10-year cutoff and retaining the six leading EOFs. The fraction of explained low-frequency variance (in %) and the ratio of low-frequency variance to total variance is given for each pattern.

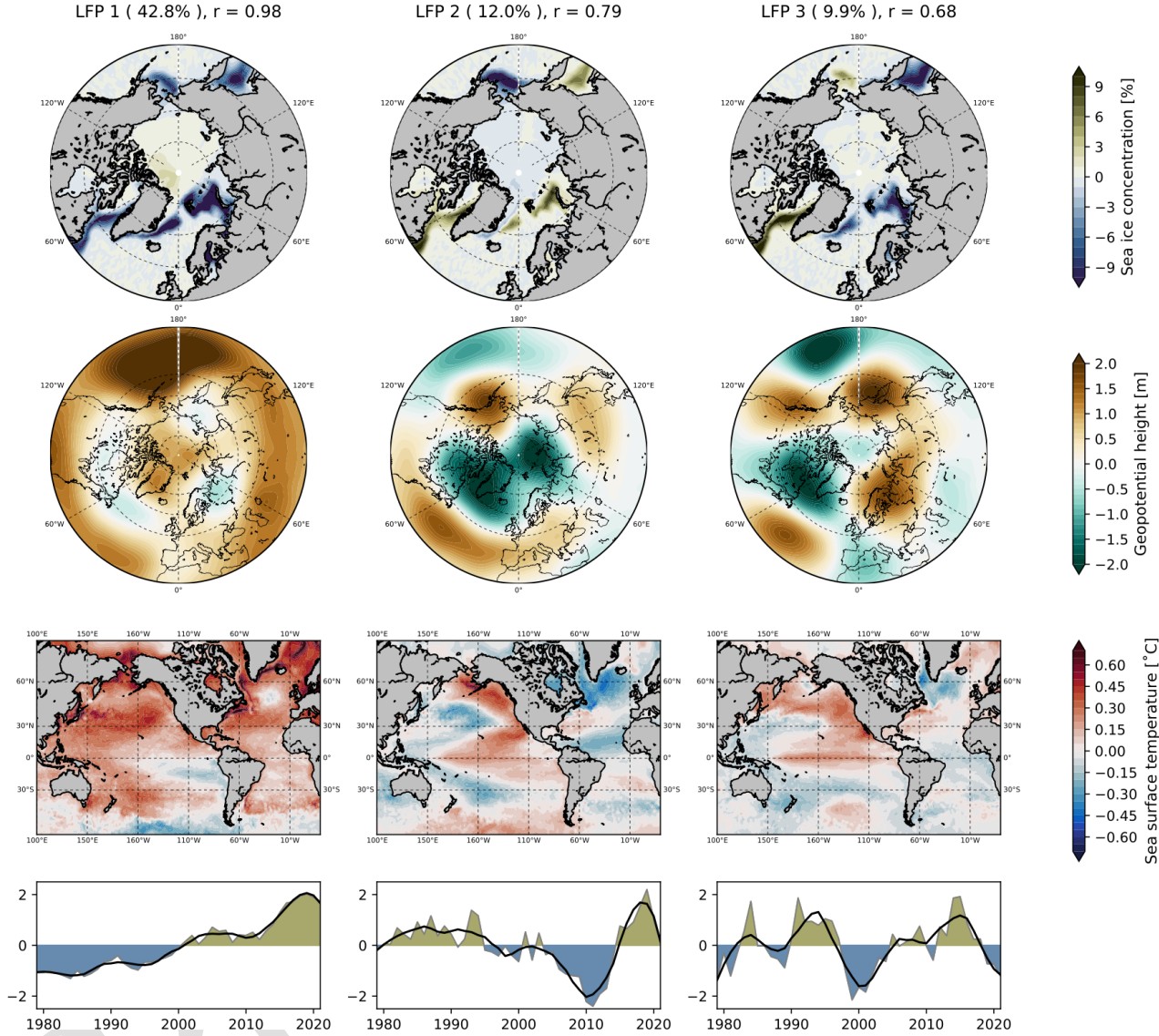

**Figure A2.** Three leading low-frequency patterns (maps) and their component time series (line plots) using winter (January–March) sea-ice concentration, winter (January–March) 500 hPa geopotential and winter-centered annual mean (July–June) sea-surface temperatures, using a 10-year cutoff and retaining the 10 leading EOFs. The fraction of explained low-frequency variance (in %) and the ratio of low-frequency variance to total variance are given for each pattern.

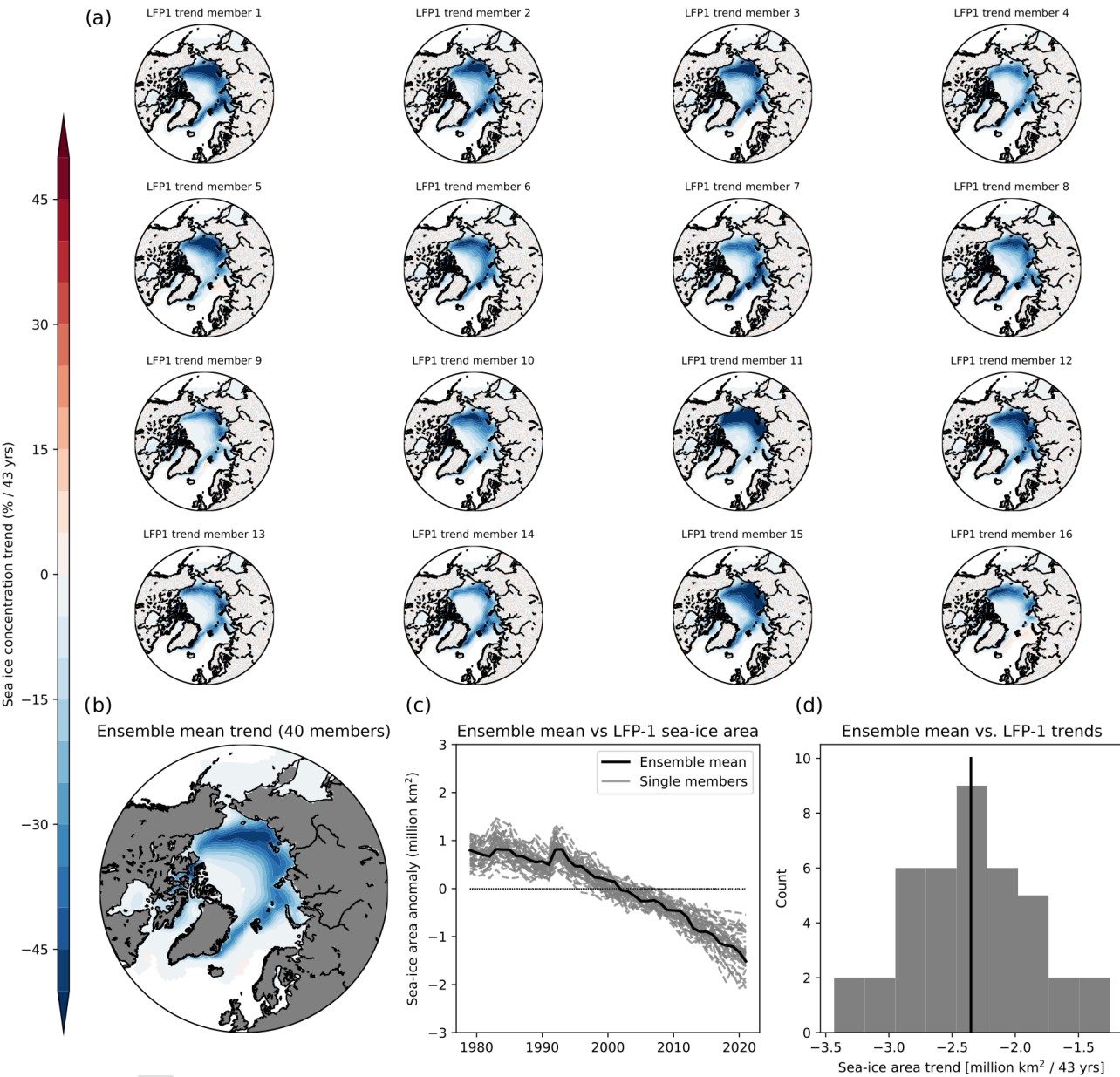

**Figure A3.** Comparing the leading low-frequency patterns and the forced summer sea-ice trends in the CESM large ensemble. **(a)** Leading low-frequency pattern using the combined analysis described in the "Materials and methods" section for the first 16 members of the CESM large ensemble (Kay et al., 2015). **(b)** Ensemble mean trend in July–September mean sea-ice concentration over all 40 members of the CESM large ensemble. **(c)** Time series of the leading low-frequency components projected onto the pan-Arctic sea-ice area for all 40 members (gray lines) and the ensemble mean pan-Arctic sea-ice area anomalies (black line). **(d)** Histogram of 43-year trends of the time series in **(c)**.

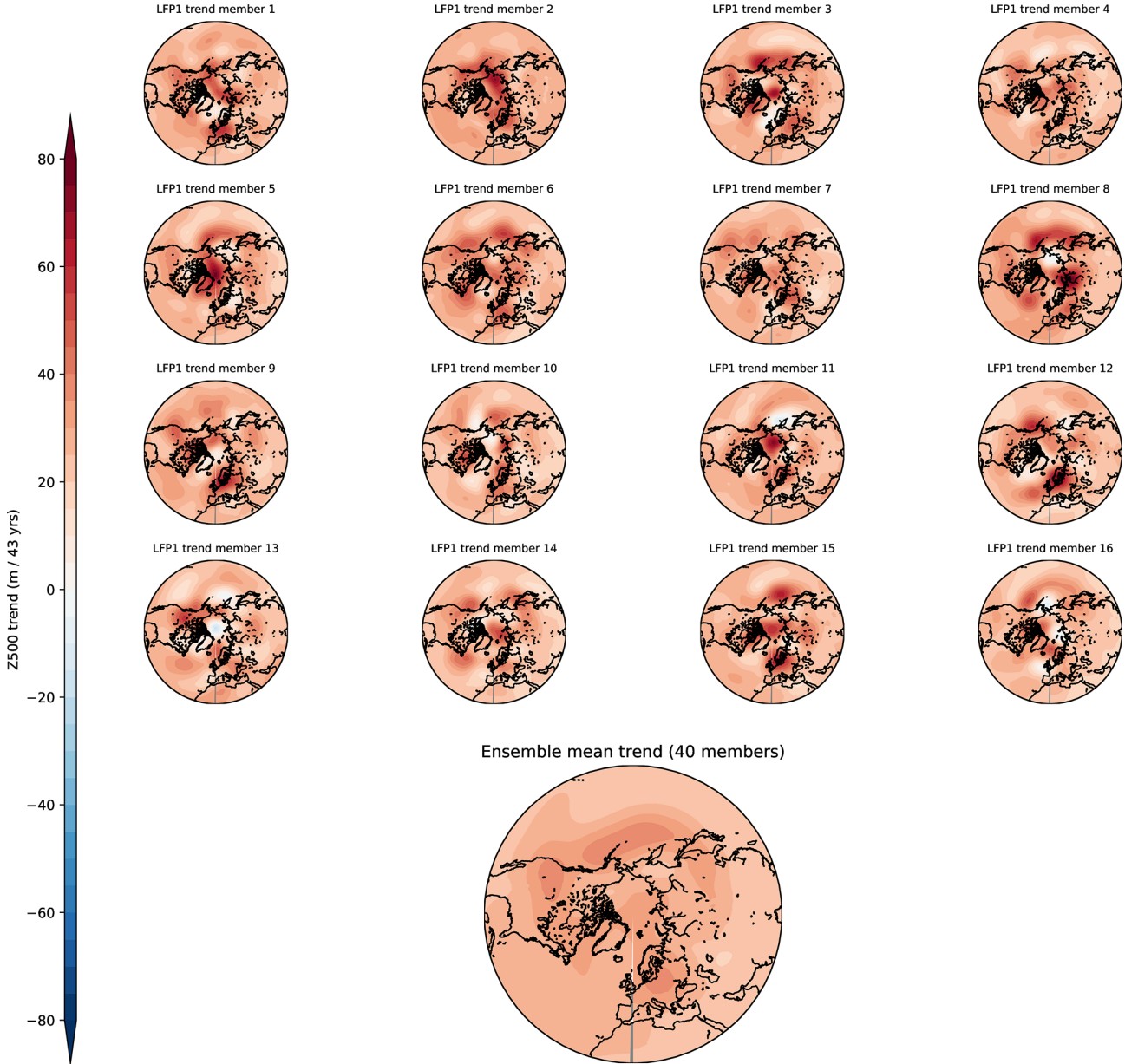

**Figure A4.** Comparing the leading low-frequency patterns and the forced summer atmospheric circulation trends in the CESM large ensemble. Same as Fig. A3a and b but for the trends in 500 hPa geopotential height (Z500).

*Data availability.* All data used in this study are freely available. OSI SAF gridded sea-ice concentration data are available at https://doi.org/10.15770/EUM_SAF_OSI_0008 (OSI SAF, 2017) and https://doi.org/10.15770/EUM_SAF_OSI_NRT_2008 (OSI SAF, 2020) TS2. The output from ERA5 is available through the Copernicus Climate Change Service: https://doi.org/10.24381/cds.f17050d7 (Hersbach et al., 2023). Monthly climate indices for the Pacific Decadal Oscillation, Arctic Oscillation and North Atlantic Oscillation can be downloaded from (NOAA, 2022a, b, c), respectively. The monthly mean North Pacific Gyre Oscillation index data can be downloaded from Di Lorenzo (2022). Output from CESM-LE is available from Climate Data Gateway (2021). Python scripts to run LFCA on OSI SAF sea-ice concentration data and produce Fig. 2 can be found at https://doi.org/10.5281/zenodo.7915287 (Dörr, 2023). LFCA is available as Python or MATLAB code under https://doi.org/10.5281/zenodo.7940013 (Wills and Shen, 2023).

*Author contributions.* JD, MÅ, DBB and RCJW conceived the study. All authors interpreted the results and were involved in reviewing and editing the text. JD performed the analysis, wrote the text and created the figures. RCJW helped with applying the method to sea ice.

*Competing interests.* The contact author has declared that none of the authors has any competing interests.

*Acknowledgements.* This study was funded by the Research Council of Norway projects Nansen Legacy (grant 276730) and the Trond Mohn Foundation (grant BFS2018TMT01). We thank Andrew Thompson for helpful discussions and comments. Furthermore, we thank Qinghua Ding and one anonymous reviewer for providing helpful comments that improved the quality of this study.

*Financial support.* This research has been supported by the Norges Forskningsråd (grant no. 276730) and the Trond Mohn Foundation (grant no. BFS2018TMT01). David B. Bonan was supported by the National Science Foundation Graduate Research Fellowship Program (NSF grant DGE-1745301). Robert C. J. Wills was supported by the National Science Foundation (NSF grant AGS-2203543). TS3

*Review statement.* This paper was edited by Bin Cheng and reviewed by two anonymous referees.

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

**Remarks from the typesetter**

TS1    Thank you for the new figures. As they differ to the previous ones; please provide an explanation why they need to be changed. We will have to ask the handling editor for their approval before we can include them. Thank you for your understanding.

TS2    Please note slight adjustments as the DOIs should also be mentioned in the data availability.

TS3    Please confirm.