# Peer review of "Forced and internal components of observed Arctic sea-ice changes"

_The Cryosphere, 2023_

## Author Response (AR1)

**Response to Reviewers**

**Note: In this response, the reviewers comments have been left untouched, and we added our responses in bold.**

Reviewer 1:

How much of Arctic sea ice melt over the past decades was due to impacts of natural processes remains very uncertain, although many studies have been devoted to addressing it by utilizing different approaches. This study represents one of these recent attempts aiming to tease apart the forced component of Arctic sea ice melt in summer and winter from those unforced parts over the past 43 years using a statistical tool known to be effective in disentangling complex variability with different origins. The method was used twice in this study on sea ice field alone and joint fields including sea ice, SST, and circulation. The identified leading modes were then carefully assessed and further linked to possible forcing mechanisms and impacts on other different local and remote fields. The main finding is that internal variability, most of which is captured by high-order modes, may explain 10% and 40% of the sea ice decline over the entire period and the 13 years (2000-2012), respectively. This finding supports some of the early studies on the same topic from the observation point of view, which is the most novel contribution of this work. Thus, this study makes some new scientific contribution to our physical understanding of the issue. Regarding the quality of its reasoning and presentation, I feel that it is a well-prepared and well-written paper with clear and well-designed figures. Taken together, my recommendation is the acceptance of the paper with minor revision. The remaining concerns I still have are that LFP 1 (Fig. 6) may still contain some signals belonging to natural variability. At least, the null-hypothesis of this argument (the leading mode has nothing to do with anthropogenic forcing) should be tested or discussed somewhere in the paper.

**Dear Prof. Ding, we thank you for your positive and thorough review of our manuscript. We agree with your assessment that combined LFP1 may still contain natural modes which are too slow to be captured by the method over the 43-year long record. We respond to the main points below.**

The Z500 pattern associated with LFP1 exhibits a strong regionality in the Arctic with a high-pressure center over Greenland, which is similar to that in LPF2 and also at odds with those forced patterns generated by models. The authors can plot the same field associated with the leading LPF in supplementary Fig. 3a. I am prtetty sure that a very uniform Z500 change will be seen. This leads me to doubt that some signals in this LPF may own a strong origin from regional circulation variations, which are usually not favored by anthropogenic forcing to the first-order approximation.

**It is true that combined LFP1 in Figure 6 shows the strongest Z500 change over Greenland, and generally stronger changes in Z500 over the mid- and high latitudes. We have checked the forced Z500 trends from CESM, and while they are much more uniform, they also show enhanced trends over the mid- and high latitudes, and also in places that agree with our LFP1 in Figure 6 (like eastern Europe and the north Pacific). Following the reviewer's suggestion, we have expanded the discussion on LFP1 and to what extent this is representative of the forced signal (lines 324-326, 400-402). We have also expanded the discussion of LFCA applied to CESM to include trends in Z500 in Lines 340-344 and Fig. A4.**

In addition, the time series of this mode (LPF1) shows a surge in 1990. I don't think this is due to a change

in CO2 concentration or aerosol forcing around the same time (aerosol forcing should favor a drop in sea ice). How to explain this feature?

**In the revised manuscrupt (lines 254-259) we now argue that LFCA likely picks up on the impact of the two volcanic eruptions of Mt. El Chichón (1982) and Mr. Pinatubo (1991) on Arctic sea ice and global temperatures, as these dips and subsequent recoveries line up with those two eruptions.**

A new test could be done to test the aforementioned null hypothesis. In the CESM LEN Pre-industrial simulation, a 43-year period with a monotonic decline or increase of Pan-Arctic sea ice can be selected, although these trends could be much weaker than the observed one over the past 43 years. The LPF method can be further used on this period to see what the leading LPF mode is. Since, by design, these sea ice trends are 100% internally driven, we would like to see how LFP can disentangle these variability when anthropogenic forcing is completely absent, but variability may also show some long-term trend signals.

**We agree that a test on the piCTRL run of CESM1 could be performed in order to test the null hypothesis. However, we believe our analysis of the full CESM1-LE ensemble is a sufficient and equivalent test, as it is a case where the forced signal is known and can be compared with what is obtained from the LFCA (LFP1) for each member. We believe that the fact that we see a very similar pattern with a downward trend in each of the 40 members, and that the temporal evolution and spatial pattern is very similar to the known forced signal in CESM1-LE is evidence enough to reject the null hypothesis that LFP1 is unrelated to the external forcing. We have expanded the presentation and discussion of the CESM1-LE analysis to make this more clear (lines 332; 337-338).**

It is nice to see that the authors examined lead-lag connections of each LPF mode with pre-season variability. I am wondering how these leading modes in summer connect with those LPF modes in winter directly.

**The summer modes generally don't correlate well with the winter modes, except for a weak correlation of +0.31 between summer LFP2 and winter LFP2 (both related to the Arctic Oscillation). We have added this in lines 211-212.**

In Fig. 4, the contour interval should be smaller for the summer case. 30m may be better to clearly illustrate the circulation pattern.

**We have changed this in Figure 4 and Figure 5.**

Reviewer 2:

Review on "Forced and internal components of observed Arctic sea-ice changes" by Dörr et al.
Internal variability of the climate system influences decadal trends in the Arctic sea-ice cover and can mask the response of sea ice to global warming. This study used Low-Frequency Component Analysis (LFCA) to separate decadal modes of variability from the observational record of summer and winter Arctic sea-ice concentration. The modes separated by LFCA account for most of the observed decadal sea ice variations in different regions of the Arctic, which can be related to mechanisms of atmospheric and oceanic variability discussed in the previous literature. Based on my knowledge, the main highlight of this work is the identification of the effects of different internal variability on sea ice changes in different decade and region, which are superimposed under the forcing of climate change and regulate the changes in Arctic sea ice at different time scales. This is an innovative work that is of great significance for understanding the mechanism of Arctic sea ice change and has the potential to be published in TC and shared by the Cryosphere and Arctic climate research communities. Therefore, I recommend that this study require a minor revision before it can be considered for publication.

**We thank the reviewer for the positive and constructive feedback. We respond to the detailed points below.**

Here are the detail comments: General: The interactions of atmosphere, sea ice and ocean within the Arctic Ocean will affect and regulate the changes and spatial distribution of sea ice through a series of feedback processes, as well as the impact of climate warming and internal variability on sea ice changes. The advection transport of Arctic sea ice will affect the spatial redistribution of sea ice mass balance (Sumata et al., 2023). The positive feedback of sea ice albedo and the shortwave radiation absorbed by the upper ocean will also adjust the response of sea ice to climate warming and internal variability (Lei et al., 2016). Although the interaction processes within the Arctic Ocean is not the focus of this study, it is helpful for understanding some statistical relationships and it is recommended to increase some discussions.

• As Arctic sea ice decreases and multi-yearice become first-years ice, the response of sea ice to atmospheric and oceanic forcing will gradually strengthen. After 2007, Arctic sea ice seems to have entered a new regime (Sumata et al., 2023), and under the influence of the same/similar intensity of internal variability of the climate system, the response of current sea ice may be different. Can this change be identified in the author's statistical analysis.

**We thank the reviewer for this interesting comment. One way we think we are seeing hints of this is some changes in the principal components in the combined summer LFP2 (Fig. 6), which shows an increase in the amplitude over time. This could indicate an increasing sensitivity of summer sea ice to atmospheric circulation in summer. However, we note that such changes in influence are likely not picked up by the patterns, since LFCA assumes that the relative amplitude of sea ice, atmospheric and SST anomalies is constant in time. We have added a short discussion on this to the manuscript in lines 267-271.**

• After 2007, the summer Arctic Ocean sea ice extent did not show a significant trend of change, but showed extremely strong interannual variability . Is this because the response of sea ice to external forcing has increased, or the internal variability of the climate system has increased?

**In the manuscript, we point out in Section 3.4 and in Section 4 that the absence of trends in the last 15 years (since ca. 2007-2012) is likely internally driven, mostly by the second combined LFP (Figure 6).**

Specially comments: • Line 15 "Arctic sea ice is declining in all seasons": it should be indicated the time period during which it occurred.

**We have modified the part to 'has declined in all seasons over the satellite record since 1979'**

**in line 16-17.**

• Line 71 "summer (July–September) and winter (January–March)": Why is the definition of summer and winter is not the conventional seasonal division in the Northern Hemisphere, but one month delay. It needs to be explained.

**We have added a sentence (lines 75-76) explaining that we want to include the sea ice minimum and maximum in the definitions (September and March), respectively.**

• Figure 1: Whether the Bering-Chukchi Sea subregion also includes the Okhotsk Sea; In addition, from the graph, it includes the Beaufort Sea, but there is inconsistency through the main text. If it includes the Beaufort Sea, the area is completely covered by ice in winter, and its changes will be affected by the coastline (there are some literatures on this issue). Therefore, the response of sea ice extent changes to climate change will also be affected by the coastline.

**We had forgotten to mention in the text that this region includes the Beaufort Sea, and thank the reviewer for spotting this. We now also mention the Beaufort Sea in line 100. Furthermore, we have added a note in lines 103-105 that from January–March, the Beaufort and Chukchi Seas are fully ice-covered, such that winter sea-ice variability in the Bering, Chukchi and Beaufort Seas region is only occurring in the Bering Sea. We agree that changes in winter sea ice in the Beaufort Sea might be affected by geometric effects from the coastlines, however, during the satellite record, this region is still fully ice-covered, such that these effects do not yet play a role.**

• Figure 4: Regression coefficients: Are they all significant? How is SST obtained in the ice zone?

**We have changed Figures 4 and 5 to show significant regions. We use SST from ERA5, which sets SSTs in ice-covered grid cells to the freezing temperature.**

• Line 177: "leading the LFCs by six months in Figure 5": If this lead time changes, will the research conclusion change or still robust?

**The patterns are similar if we decrease/increase lead time by +/- two months. If we change the lead time more than that, the target periods change (i.e. the previous winter becomes the previous fall, or spring), and then the results look different, as expected.**

• Line 311: "Slow modes of internal variability" What is the meaning for the slow modes here.

**We have specified 'with a time scale of 25 years of more' in line 330-331.**

• Line 345 "However, summer sea ice will disappear in the coming decades (Notz and Community, 2020; Årthun et al., 2020; Bonan et al., 2021b), setting a limit to the length of the observational record of sea ice." I think you can delete this sentence: (1) There is controversy over the occurrence of ice free phenomena in the Arctic Ocean during summer, and it is not certain that will occur. 2) Even if there is no ice, it is short-lived. On average, there will still be sea ice during summer, and the situation is change every year, so there will still be interannual changes.

**We believe the sentence is justified, because (1) we are not aware of any scientific controversy over the occurrence of ice free summers around the middle of this century, and (2) while that may be true for an optimistic emissions scenario, we believe even single summers without ice cover could lead to artefacts in the method. We have, however, modified the sentence somewhat to reflect that this statement about the future is based on models. In line 369, we now write "summer sea ice is projected to disappear...".**

• The Low-Frequency Component Analysis (LFCA) is a new method. Thus, for the convenience of promotion, I strongly recommend that the authors can share relevant calculation code, at least for the code for the

LFCs of interdecadal changes in Arctic sea ice concentration.

**This is a good point. In lines 414-416, we have added a link to a github repository containing matlab and python code for LFCA, and link to a repository with the code that performs LFCA on sea ice and creates Figure 2.**

Reference: Sumata, H. et al., Regime shift in Arctic Ocean sea ice thickness, Nature, 2023, https://doi.org/10.1038/s41586-022-05686-x. Lei, R. et al. Changes in summer sea ice, albedo, and portioning of surface solar radiation in the Pacific sector of Arctic Ocean during 1982–2009. J. Geophys. Res. Oceans 121, 5470–5486 (2016).

---

## Author Response (AR2)

**Response to Reviewers**

**Note: In this response, the reviewers comments have been left untouched, and we added our responses in bold.**

Reviewer 2:

This paper has been made corresponding revisions based on the previous round of review. I have no further substantive comments here, only some specific suggestions:

**We thank the reviewer for the helpful suggestions. We respond to them below.**

1)The results of this study are all based on relatively short (40 years) satellite remote sensing data. Although the author believes that the low-frequency signal recognition method used can reduce the impact of short time series, I consider it is still necessary to further discuss some results that may be affected by insufficient time series, and its possible degree or extent or intensity.
**We thank the reviewer for this suggestion. We already discuss the limitations of using this method on a short time series in Lines 320–345 and Lines 366–375. We have furthermore added the following sentence to Line 370-371: 'The results may therefore overestimate the forced contribution to sea ice changes, and underestimate the internal contribution.'**

2)Figure 1 "sea ice edge based on 50% sea ice concentration" : Why do you use a 50% threshold instead of the usual 15%?

**Since this figure is showing seasonal average sea ice cover, we think it is more useful to use a 50% threshold to not skew the cover to one month. In the figure caption, we have changed 'sea-ice edge' to 'sea-ice cover' to more accurately describe what we are showing.**

3)Figure 2b: Corresponding to LFP2 in summer, Beaufort Sea and East Siberian Sea have an obvious dipole structure, which needs further explanation. It may be related to the leading (some months) Beaufort High, which can adjust the zonal advection of sea ice in the southern Beaufort Sea Chukchi Seas.

**Yes, there is a dipole structure, and this ties into the discussion on the effect of the previous winter AO on summer sea ice through ice motion and subsequent thinning. We have added a sentence in Line 196-197, specifically mentioning that this also leads to a dipole between the Beaufort and East Siberian Seas.**

4)Line 199 "The third LFP in summer is connected to a dipole in geopotential height over western Greenland and the Canadian Archipelago on the one side, and the Nordic Seas on the other side": Can this be considered as an Arctic dipole, the so-called DA.

**Yes, it is reminiscent of that. We have added a sentence, and some references in Lines 202–203.**

5)Line 369 "However, summer sea ice is projected to disappear in the coming decades (Notz and SIMIP community, 2020; Årthun et al., 2020; Bonan et al., 2021b), setting a limit to the length of the observational record of summer sea ice": The threshold for defining the Arctic ice free is 1 million square kilometers by the climate models, and it can be considered as ice free with a temporary appearance, not for the whole summer. Thus, interannual changes in summer sea ice extent will still exist. In the time scale of the next 50-70 years, there should be a pseudo-proposition of "no statistical change of summer sea ice because of ice free". This proposition may only be established in some very outer sea areas, such as the Barents Sea and Baffin Bay.

**We have removed the last part of the sentence and replaced 'summer' with 'September' in Line 372. We now use the same argument than for winter, that patterns may change over time.**